# Does Bladder Cancer with Inchworm Sign Indicate Better Prognosis after TURBT?

**DOI:** 10.3390/cancers14235767

**Published:** 2022-11-23

**Authors:** Ryunosuke Nakagawa, Kouji Izumi, Renato Naito, Suguru Kadomoto, Hiroaki Iwamoto, Hiroshi Yaegashi, Shohei Kawaguchi, Takahiro Nohara, Kazuyoshi Shigehara, Kotaro Yoshida, Yoshifumi Kadono, Atsushi Mizokami

**Affiliations:** 1Department of Integrative Cancer Therapy and Urology, Kanazawa University Graduate School of Medical Science, Kanazawa 920-8641, Japan; 2Department of Radiology, Kanazawa University Graduate School of Medical Science, Kanazawa 920-8641, Japan

**Keywords:** inchworm sign, non-invasive muscle bladder cancer, muscle invasive bladder cancer, bladder neck, non-papillary tumor

## Abstract

**Simple Summary:**

Inchworm sign is considered to be a characteristic finding in non-muscle invasive bladder cancer. We retrospectively investigated the factors related to muscle invasive status in bladder cancer associated with inchworm sign and the role of inchworm sign in tumor outcomes following transurethral resection of bladder tumor. Of the 109 patients with inchworm sign, 94 (86.2%) were non-muscle invasive bladder cancer. Non-papillary tumors and tumors located in the bladder neck were significant predictors of muscle invasive bladder cancer with inchworm sign. Additionally, inchworm sign was not a prognostic factor in patients with non-muscle invasive bladder cancer in this study.

**Abstract:**

Background: Inchworm sign is considered to be a characteristic finding in non-muscle invasive bladder cancer (NMIBC). Nevertheless, pathologically diagnosed muscle invasive bladder cancers (MIBCs) are occasionally diagnosed from tissue obtained by transurethral resection of bladder tumor (TURBT) in patients with inchworm sign. Methods: We retrospectively investigated the factors related to muscle invasive status in bladder cancer associated with inchworm sign and the role of inchworm sign in tumor outcomes following TURBT. Results: Of the 109 patients with inchworm sign, 94 (86.2%) and 15 (13.8%) were NMIBC and MIBC, respectively. Non-papillary tumors (hazard ratio (HR): 9.55, 95% confidence interval (CI): 2.07–44.10; *p* < 0.01) and tumors located in the bladder neck (HR: 7.73, 95% CI: 1.83–32.76; *p* < 0.01) were significant predictors of MIBC in bladder cancer with inchworm sign. Furthermore, recurrence-free survival (RFS) and progression-free survival were compared between patients with NMIBC with and without inchworm sign; however, no significant differences were found. In patients with NMIBC with inchworm sign, positive urine cytology was a prognostic factor for RFS (HR: 1.90, 95% CI: 1.04–3.48; *p* = 0.04). Conclusions: In bladder cancer with inchworm sign, 86.2% were NMIBC. Even in the case of inchworm sign, the presence of a non-papillary tumor or a bladder neck tumor before TURBT should be noted because of the possibility of MIBC. In this study, the inchworm sign was not a prognostic factor in patients with NMIBC.

## 1. Introduction

Bladder cancer (BC) is the tenth most commonly diagnosed cancer, and the age-standardized incidence rate (per 100,000 person-years) is 9.5 for men and 2.4 for women worldwide [1]. Treatment for BC depends on the stage of the disease; however, the degree of cancer invasion is one of the most important factors [2]. While total cystectomy is the standard treatment for BC that has invaded the muscle layer, which is called muscle invasive bladder cancer (MIBC), non-muscle invasive bladder cancer (NMIBC) is likely to be treated by follow-up or bladder-preservation options, such as Bacille Calmette–Guerin (BCG), depending on the risk [3]. Transurethral resection of bladder tumor (TURBT) is commonly performed for BC not only for therapeutic but also for diagnostic purposes, and the pathological results are used to determine treatment options [2,3,4]. Moreover, it is useful for predicting cancer invasion by preoperative imaging evaluation. Panebianco et al. established an imaging protocol using multi-parametric magnetic resonance imaging (MRI), called VI-RADS, to classify and evaluate the risk of muscle invasion into five categories [5]. Underdiagnosis by TURBT is often a challenge [6]; however, standardization of imaging evaluation and prediction of muscle layer invasion may prevent it.

Diffusion-weighted imaging (DWI) is constructed by quantifying the diffusion of the water molecules in tissues and is excellent in the delineation of BC [7], and the signal reflects the characteristics of the tumor tissue [8]. DWI is more accurate than T2-weighted MRI in BC staging, particularly in staging-localized lesions of pT1–2 [9]. The “inchworm sign” in DWI, proposed by Takeuchi et al., is reported to predict NMIBC [10]. They suggested that inchworm sign showed a bow-shaped tumor with high signal intensity and a submucosal stalk with low signal intensity on DWI. They also reported that the inchworm sign was a characteristic finding in all 28 pT1 cases studied and was useful in differentiating MIBC.

In this study, we focused on the inchworm sign and investigated the factors related to muscle invasive status in BC with inchworm sign and the role of the inchworm sign in tumor outcomes following TURBT.

## 2. Patients and Methods

### 2.1. Patient Selection and Collection of Clinical Data

Among the patients with BC receiving treatment at Kanazawa University Hospital from 2007 to 2019, those with BC without metastasis and with inchworm sign on DWI were included in this study. Only initial cases were included, and recurrent cases were not included. Cases with multiple tumors were included. Patients who did not have an MRI at diagnosis or had an MRI but no DWI were excluded. We also excluded cases with coexisting upper tract urothelial carcinoma at the initial diagnosis. The probability of NMIBC and predictors of MIBC in these patients with BC were retrospectively investigated. The inchworm sign is defined as a finding on diffusion-weighted (DW)-MRI that shows a bow-shaped tumor with high signal intensity and a submucosal stalk with low signal intensity.

Moreover, patients with NMIBC who had an MRI at diagnosis were selected and categorized into two groups with and without inchworm sign. The recurrence-free survival (RFS) and progression-free survival (PFS) of both groups were compared. RFS was measured from the diagnosis of BC to recurrence or last follow-up. PFS was measured from the diagnosis of BC until recurrence of MIBC or appearance of metastasis or last follow-up. Follow-up was terminated on March 2022. Clinical stage was determined based on the eighth edition of the Union for International Cancer Control Tumor, Node, and Metastasis classification published on 2017.

### 2.2. Image Analysis

A single urologist with 7 years of experience in reading abdominal DWI data and performing TURBT (Reviewer 1) reviewed the MR images blinded to the clinical and histologic information. The tumor image with high signal intensity and a submucosal stalk with low signal intensity on DWI were identified as positive for the inchworm sign. To ensure interobserver concordance in assessment of the inchworm sign, one reviewer (Reviewer 2—a radiologist with 19 years of experience reading abdominal DWI data), who did not disclose the clinical course and histological findings, independently evaluated the presence of inchworm signs on all images.

### 2.3. Statistical Analyses

Binomial logistic regression analysis was used to evaluate the predictive impact of potential factors of MIBC in patients with positive inchworm sign. The Cox proportional hazards model was used to evaluate the predictive impact of potential factors of RFS and PFS. All factors were divided into dichotomized variable categories and evaluated. Hazard ratios (HRs) and 95% confidence intervals (CIs) were calculated. RFS and PFS were estimated using the Kaplan–Meier method, with differences being compared using log-rank tests. Statistical analyses were performed using the commercially available software Prism 8 (GraphPad, San Diego, CA, USA) and SPSS ver. 25.0 (SPSS Inc, Chicago, IL, USA), with *p* values of <0.05 indicating statistical significance.

### 2.4. Ethical Considerations

This study was approved by the Institutional Review Board of Kanazawa University Hospital (2021-300). Informed consent was obtained in the form of opt-out posted at our facility allowed by the Medical Ethics Committee of Kanazawa University. All methods were performed in accordance with the relevant guidelines and regulations.

## 3. Results

### 3.1. Patient Characteristics with Inchworm Sign

A representative inchworm sign, showing a bow-shaped tumor with high signal intensity and a submucosal stalk with low signal intensity on DW-MRI, is shown in Figure 1.

The characteristics of the patients with BC with inchworm sign on DW-MRI are presented in Table 1. A total of 109 patients were identified with inchworm sign. Regarding pT stage, 2 (1.8%), 48 (44.0%), 44 (40.4%), and 14 (13.8%) patients were Tis, Ta, T1, and T2 and above, respectively. A total of 94 (86.2%) patients had NMIBC. Additionally, 38 (34.9%), 92 (84.4%), 46 (42.2%), 82 (75.2%), and 73 (67.0%) patients had a tumor size of >30 mm, papillary tumors, multiple tumors, hematuria, and positive urine cytology, respectively. The most frequent tumor location was the lateral wall (54.1%). There was a total of 37 surgeons in the 12-year period, due to staff transitions at our facility. All surgeons were similar in terms of surgical techniques. For surgical instruments, an Olympus TURis bipolar resectoscope (Olympus Corporation, Tokyo, Japan) was used, and an Aladuck LS-DLED (SBI Pharma, Tokyo, Japan) or Storz PDD system and Storz bipolar resectoscope (KARL STORZ, Tuttlingen, Germany) was used for photodynamic diagnosis (PDD)-TURBT. Twenty-one patients (19.3%) received immediate postoperative intravesical chemotherapy. The drugs administered varied from pirarubicin, mitomycin C, and epirubicin hydrochloride.

### 3.2. Predictors of MIBC in Patients with Inchworm Sign

We examined the predictors of MIBC in patients with inchworm sign. In univariate analysis, multiple tumors (HR: 3.22, 95% CI: 1.02–10.19; *p* = 0.04), tumor size >30 mm (HR: 6.82, 95% CI: 1.99–23.31; *p* < 0.01), non-papillary tumors (HR: 5.03, 95% CI: 1.50–16.86; *p* < 0.01), and location in the bladder neck (HR: 10.79, 95% CI: 3.17–36.75; *p* < 0.01) were predictive factors for MIBC (Table 2). In multivariable analysis, non-papillary tumors (HR: 9.55, 95% CI: 2.07–44.10; *p* < 0.01) and localization to the bladder neck (HR: 7.73, 95% CI: 1.83–32.76; *p* < 0.01) were significant predictive factors for MIBC. The cases with non-papillary tumor (Figure 2a–c) and bladder neck tumor (Figure 2d–f) with inchworm sign but which are MIBC are shown in Figure 2.

### 3.3. RFS and PFS Stratified by the Presence of Inchworm Sign on Patients with NMIBC

Subsequently, patients with NMIBC (Ta and T1) with and without inchworm sign were grouped. A total of 77 patients with NMIBC without inchworm sign and 92 with inchworm sign were noted. The characteristics of patients with NMIBC are shown in Table 3. NMIBC with inchworm sign had significantly higher percentages of a tumor size >30 mm (*p* < 0.01), positive cytology (*p* = 0.03), and pT1 (*p* = 0.01). Comparison of RFS (HR: 1.49, 95% CI: 0.98–2.27; *p* = 0.06) and PFS (HR: 1.92, 95% CI: 0.62–5.96; *p* = 0.28) between the two groups showed no significant difference (Figure 3). Furthermore, we compared RFS and PFS for each patient with pTa and pT1, respectively, and found no significant differences (Figure 4).

### 3.4. Prognostic Factors for RFS and PFS of Patients with NMIBC with/without Inchworm Sign

We focused on patients with NMIBC with inchworm sign and investigated the prognostic factors for RFS and PFS. We identified positive cytology (HR: 1.90, 95% CI: 1.04–3.48; *p* = 0.04) as a predictor for RFS (Table 4). Conversely, no prognostic factors for PFS were found (Table 4). Furthermore, prognostic factors for RFS and PFS in patients with NMIBC without inchworm sign were investigated. However, positive cytology (HR: 1.60, 95% CI: 0.80–3.22; *p* = 0.19) was not a predictor for RFS, and only multiple tumors (HR: 3.04, 95% CI: 1.56–5.90; *p* > 0.01) was the prognostic factor (Table 5). No prognostic factors for PFS were noted (Table 5).

### 3.5. Interobserver Variability in Identification of Inchworm Sign

We assessed interobserver variability in the identification of inchworm signs. Reviewers 1 and 2 independently analyzed the DW images of all eligible tumors and found inchworm signs in 108 (33.2%) and 112 (34.5%) tumors, respectively. Interobserver variability between the two readers was generally excellent, with a kappa coefficient of 0.85.

## 4. Discussion

In this study, 86.2% of patients with BC with inchworm sign were NMIBC. Saito et al. reported that the stalk extending from the bladder wall to the tumor center is composed of fibrous tissue, capillaries, inflammatory cells, and edema [11]. Takeuchi et al. also reported that pT1 bladder cancer with inchworm sign histologically has a submucosal stalk consisting of significantly edematous submucosa, fibrous tissue, capillaries, and mild inflammatory cell infiltration, which is consistent with a low signal intensity region on DW-MRI, and is one of the indicators in the imaging findings of NMIBC [10]. Although there are no reports on the formation of the inchworm sign, we suspect that the thickening of the submucosa due to inflammation and edema results in the formation of this bow-shaped structure. The probability that BC with inchworm sign indicates NMIBC is high and is considered to be useful in differentiating from MIBC. However, in our study, not all patients with BC with inchworm sign had NMIBC, and 13.8% were diagnosed with MIBC. Kobayashi et al. reported that 25% of patients diagnosed with NMIBC based on the findings of inchworm sign were consequently pathologically diagnosed with MIBC [12]. They suggested that the fading images of surrounding tissues in DW-MRI may result in loss of anatomical information and compromise the staging performance, and they reported that incorporating quantitative information on apparent diffusion coefficient (ADC) values improved the accuracy of inchworm sign staging and greatly reduced the misdiagnosis rate.

We investigated patients with pathologically diagnosed MIBC despite the presence of inchworm sign and found that non-papillary tumor and localization to the bladder neck were predictive factors for MIBC by multivariable analysis. Non-papillary tumors, including nodular tumors, tend to be more malignant and have more muscle layer invasion than papillary tumors [13]. Notably, if the cystoscopy shows a non-papillary tumor, MIBC is highly likely, regardless of the presence or absence of inchworm sign. The trigone muscle presents from the deltoid to the proximal urethra, which may have functional implications for understanding the mechanisms of continence at the bladder neck [14], and the muscular layer gradually thickens from the deltoid to the proximal urethra, while the submucosal tissue becomes occupied and thin [15]. In other words, the bladder neck has a higher percentage of muscle layers. Xiao et al. reported that bladder neck tumors have significantly more muscle layer invasion and an advanced tumor stage at diagnosis, lymphovascular invasion, and higher frequency of local and distant metastases than non-bladder neck tumors [16]. We suggest that MIBC may not be ruled out even if the inchworm sign is observed.

In our study, we found no significant difference in RFS and PFS in patients with NMIBC with or without inchworm sign. Huan-Jun et al. reported that the median diameter of tumors with stems (21.5 mm) was significantly larger than that of tumors without stems (13.0 mm) [17]. They also postulated that submucosal enhancement and stalks are related to the size of the tumors and small tumors may not generate enough pulling force to form stalks. In our study, this may be related to the significantly larger tumor size in NMIBC patients with inchworm sign. Since the percentage of T1 was significantly higher in patients with inchworm sign, Ta and T1 were examined separately; however, no significant difference was observed. Yajima et al. reported that patients with T1 without the inchworm sign had a significantly lower PFS than those with the inchworm sign [18]. Conversely, they also reported no significant difference in RFS. Several reports have suggested that the degree of tumor invasion into the submucosal lamina propria is associated with the progression [19,20,21], and they proposed that the inchworm sign is one of the indicators that can easily assess the degree of invasion. The reason for the difference between our results and those of Yajima et al. is the lower BCG treatment and the second TURBT (6 of 63 cases, data are not shown) rates in our patients. These factors may impact patient prognosis. Furthermore, we identified a higher rate of positive cytology in patients with NMIBC with inchworm sign, and this factor is strongly associated with worse RFS. Jancke et al. found that a high-grade malignant bladder wash cytology at the primary diagnosis was predictive of recurrence and suggested that this could be considered in designing future follow-up schedules [22]. Furthermore, Kiyoshima et al. reported that patients with BC with a positive cytology had larger tumor diameters and that patients with a positive cytology tended to have shorter RFS [23]. However, in our study, positive cytology was a prognostic factor for worse RFS only in patients with inchworm sign; additionally, tumor size was not a significant factor regardless of the presence or absence of inchworm sign. We hypothesize that this result was due to the surface area of the tumor. A submucosal stalk with low signal intensity on DW-MRI in the inchworm sign indicates a stalked tumor; the surface area of a tumor with an inchworm sign may be larger than that of a tumor without an inchworm sign with the same length (Figure 5). This difference in surface area may have affected the higher positive urinary cytology rate in patients with inchworm sign and, consequently, may have been a prognostic factor for RFS. In addition, cancer cells in BC with the inchworm sign may have a higher tendency to segregate from the tumor. These structural differences may also explain why patients with pTa with inchworm sign tended to have worse RFS. The surface area of tumors with inchworm sign is an issue that requires further investigation.

This study has several limitations. First, this study was retrospective, and the evaluation of treatment effects was left to individual physicians, which may have resulted in bias. Second, although the second TURBT was recommended for T1 tumors, it was performed in a few patients in this study. Third, we focused on the DWI inchworm sign in this study and did not examine other imaging methods, including an ADC map. Additional imaging methods in MRI may yield different results. Finally, BCG maintenance therapy was not administered to most patients. These limitations should be considered when interpreting the results. Further studies are required to accumulate more cases.

## 5. Conclusions

The inchworm sign in BC predicted NMIBC with a high probability of 86.2%. In BC with inchworm sign, non-papillary tumors and tumors located in the bladder neck were predictors of MIBC. Although the presence of this sign indicated a high rate of NMIBC, caution should be exercised when these two factors are present. Additionally, the inchworm sign was not a prognostic factor in patients with NMIBC in this study.

## Figures and Tables

**Figure 1 cancers-14-05767-f001:**
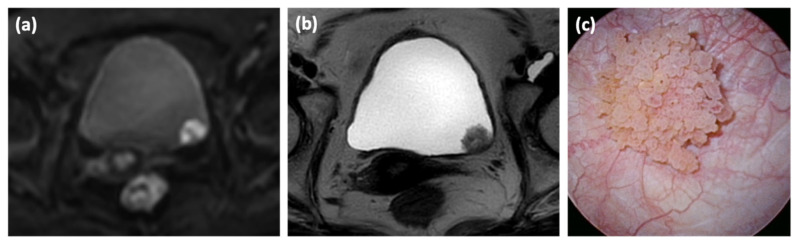
The case of bladder cancer with inchworm sign. Findings of bladder cancer in a 72-year-old woman diagnosed with hematuria. (**a**) Diffusion-weighted (DW)-MRI shows high signal intensity from the tumor and low signal intensity from the submucosal stalk. This finding is defined as the inchworm sign. (**b**) Transverse T2-weighted MRI shows a tumor on the left lateral wall of the bladder; however, no change in the internal signal intensity is observed. (**c**) Cystoscopy reveals a papillary tumor at the same site.

**Figure 2 cancers-14-05767-f002:**
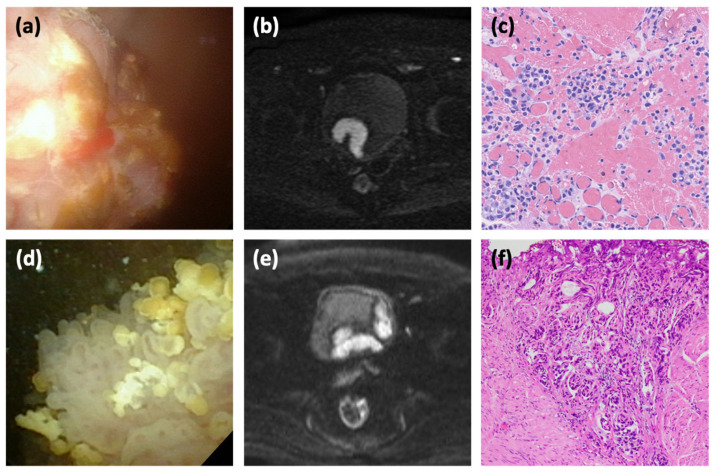
Bladder cancer with muscle invasion despite the presence of inchworm sign. A 77-year-old man consulted for hematuria (**a**–**c**). Cystoscopy shows a non-papillary tumor on the right wall (**a**); DW-MRI shows the inchworm sign (**b**); and TURBT shows tumor invasion into the muscle layer, and the diagnosis of MIBC was made (**c**). A 76-year-old man referred for urinary frequency and gross hematuria (**d**–**f**). Cystoscopy reveals a papillary tumor in the neck (**d**). MRI shows a bladder tumor with inchworm sign and multiple tumors on the lateral wall of the bladder (**e**). The resected tissue of the neck tumor shows findings of muscle invasion (**f**).

**Figure 3 cancers-14-05767-f003:**
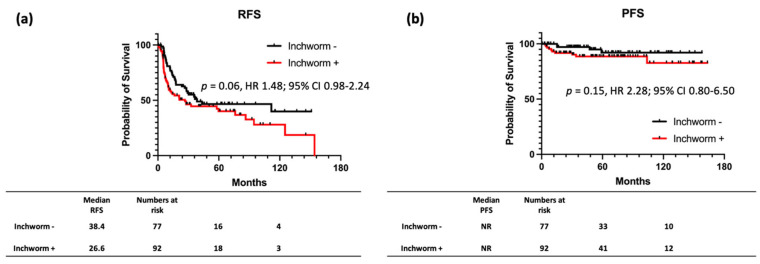
Kaplan-Meier survival curves for recurrence-free survival. (**a**) Progression−free survival. (**b**) Stratified by the presence of inchworm sign for patients with NMIBC. Comparison of RFS (HR: 1.49, 95% CI: 0.98–2.27; *p* = 0.06) and PFS (HR: 1.92, 95% CI: 0.62–5.96; *p* = 0.28) between the negative and positive inchworm sign groups shows no significant difference.

**Figure 4 cancers-14-05767-f004:**
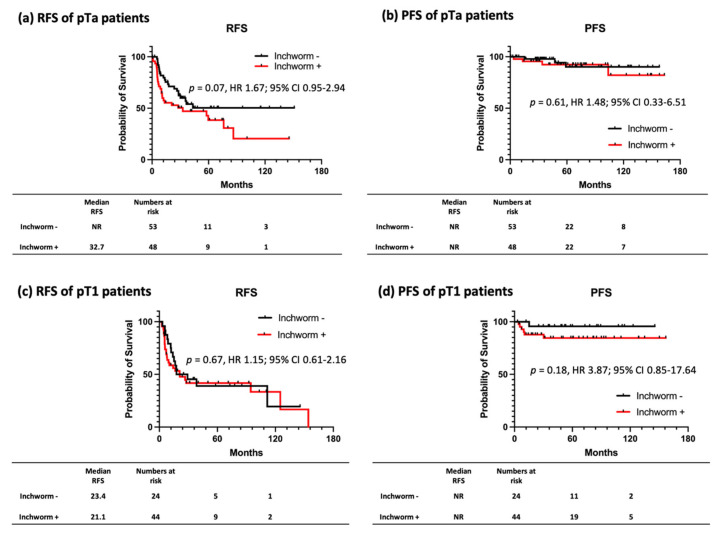
Kaplan-Meier survival curves for recurrence-free survival and progression-free survival stratified by the presence of an inchworm sign for patients with pTa and pT1. Patients with NMIBC are compared for RFS and PFS with and without inchworm sign in pTa and pT1. In patients with pTa, RFS tends to be worse in those with inchworm sign; however, the difference is not significant (**a**). No other comparisons are significantly different (**b**–**d**).

**Figure 5 cancers-14-05767-f005:**
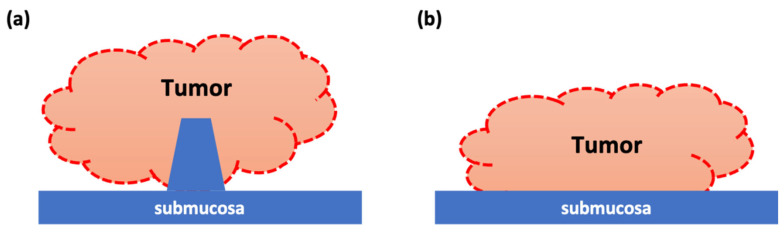
Schematic diagram of bladder cancer. Bladder cancer (BC) with inchworm sign has a submucosal stalk (**a**), whereas BC without inchworm sign does not (**b**). Therefore, for tumors of the same size, the surface area of the tumor protruding into the bladder lumen is larger in BC with inchworm sign (red dashed line).

**Table 1 cancers-14-05767-t001:** Characteristics of the patients with inchworm sign.

Variable		
No. of patients		109
Age (median)		71
Sex		
	Male (%)	90 (82.6)
	Female (%)	19 (17.4)
Histology		
	UC (%)	94 (86.2)
	UC with squamous differentiation (%)	10 (9.2)
	UC with glandular differentiation (%)	5 (4.6)
Pathological grade		
	1 (%)	4 (3.7)
	2 (%)	65 (59.6)
	3 (%)	30 (27.5)
	Unknown (%)	10 (9.2)
pT stage		
	Tis (%)	2 (1.8)
	Ta (%)	48 (44.0)
	T1 (%)	44 (40.4)
	T2 (%)	11 (10.1)
	T3 (%)	2 (1.8)
	T4 (%)	2 (1.8)
Tumor size		
	10–30 mm (%)	71 (65.1)
	30 mm < (%)	38 (34.9)
Tumor location		
	Anterior wall (%)	13 (11.9)
	Posterior wall (%)	17 (15.6)
	Side wall (%)	59 (54.1)
	Dome (%)	3 (2.8)
	Neck (%)	17 (15.6)
Papillary		
	Yes (%)	92 (84.4)
	No (%)	17 (15.6)
Multiple tumors		
	Yes (%)	46 (42.2)
	No (%)	63 (57.8)
Hematuria		
	Yes (%)	82 (75.2)
	No (%)	26 (23.9)
	Unknown (%)	1 (0.9)
Cytology		
	Negative (%)	36 (33.0)
	Positive (%)	73 (67.0)

**Table 2 cancers-14-05767-t002:** Predictors of MIBC in patients with positive inchworm sign.

Variables	Univariate				Multivariable			
	95% CI				95% CI			
	*p* Value	HR	Lower	Upper	*p* Value	HR	Lower	Upper
Age ≥ 75 (years)	0.49	1.48	0.49	4.42				
Female	0.72	0.75	0.15	3.65				
Cytology positive	0.15	3.12	0.65	14.89				
Hematuria	0.65	1.37	0.36	5.28				
Multiple tumors	0.04	3.22	1.02	10.19	0.09	3.38	0.83	13.69
Tumor size >30 mm	<0.01	6.82	1.99	23.31	0.053	4.27	0.98	18.64
Non-papillary tumor	<0.01	5.03	1.50	16.86	<0.01	9.55	2.07	44.10
Bladder neck	<0.01	10.79	3.17	36.75	<0.01	7.73	1.83	32.76

**Table 3 cancers-14-05767-t003:** Characteristics of patients with NMIBC.

Variable		Inchworm Sign −	Inchworm Sign +	*p* Value
No. of patients		77	92	
Age (median)		73	71	0.35
Sex				0.75
	Male (%)	65 (84.4)	76 (82.6)	
	Female (%)	12 (15.6)	16 (17.4)	
Tumor size				<0.01
	10–30 mm (%)	72 (98.7)	65 (64.9)	
	>30 mm (%)	5 (1.3)	27 (35.1)	
Multiple tumors				0.42
	Yes (%)	34 (44.2)	35 (38.0)	
	No (%)	43 (55.8)	57 (62.0)	
Histology				0.66
	UC (%)	71 (92.2)	81 (88.0)	
	UC with squamous differentiation (%)	2 (2.6)	4 (4.3)	
	UC with glandular differentiation (%)	4 (5.2)	7 (8.7)	
Pathological grade				0.11
	1 (%)	9 (11.7)	4 (4.3)	
	2 (%)	45 (58.4)	61 (66.3)	
	3 (%)	11 (14.3)	19 (20.7)	
	Unknown (%)	12 (15.6)	8 (8.7)	
Cytology				0.03
	Negative (%)	38 (49.4)	34 (37.0)	
	Positive (%)	33 (42.9)	56 (60.9)	
	Unknown (%)	6 (16.5)	2 (2.1)	
pT stage				0.01
	Ta (%)	53 (68.8)	44 (47.8)	
	T1 (%)	24 (31.2)	48 (52.2)	
BCG treatment				0.39
	Yes (%)	15 (19.5)	23 (25.0)	
	No (%)	62 (80.5)	69 (75.0)	
PDD				0.13
	Yes (%)	3 (3.9)	9 (9.8)	
	No (%)	74 (96.1)	82 (90.2)	

**Table 4 cancers-14-05767-t004:** Univariate analysis of the prognostic factors for recurrence-free survival and progression-free survival of patients with NMIBC with positive inchworm sign.

(a) RFS					(b) PFS				
Variables	*p* Value	HR	95% CI	Upper	Variables	*p* Value	HR	95% CI	Upper
			Lower					Lower	
Age ≥ 75 (years)	0.39	0.78	0.44	1.38	Age ≥ 75 (years)	0.72	1.27	0.36	4.53
Male	0.67	1.18	0.55	2.52	Male	0.43	2.29	0.29	18.09
Tumor size ≥ 30 mm	0.81	1.08	0.6	1.93	Tumor size ≥ 30 mm	0.15	2.51	0.73	8.72
Bladder neck	0.92	0.95	0.38	2.4	Bladder neck	0.5	0.04	>0.01	440.34
Non-papillary tumor	0.41	0.61	0.19	1.97	Non-papillary tumor	0.52	0.04	>0.01	573.38
Multiple tumors	0.29	0.74	0.42	1.3	Multiple tumors	0.58	0.68	0.18	2.64
Hematuria	0.66	1.15	0.61	2.17	Hematuria	0.25	3.38	0.43	26.8
Positive cytology	0.04	1.9	1.04	3.48	Positive cytology	0.15	47.15	0.25	9005.87
BCG treatment	0.82	0.93	0.48	1.8	BCG treatment	0.16	2.48	0.7	8.8
PDD	0.27	0.45	0.11	1.86	PDD	0.58	0.04	>0.01	2761.93

**Table 5 cancers-14-05767-t005:** Univariate analysis of the prognostic factors for recurrence-free survival and progression-free survival of patients with NMIBC with negative inchworm sign.

(a) RFS					(b) PFS				
Variables	*p* Value	HR	95% CI	Upper	Variables	*p* Value	HR	95% CI	Upper
			Lower					Lower	
Age ≥ 75 (years)	0.13	1.64	0.86	3.14	Age ≥ 75 (years)	0.65	0.6	0.06	5.71
Male	0.28	1.78	0.63	5.07	Male	0.7	0.64	0.07	6.27
Tumor size ≥ 30 mm	0.09	2.83	0.85	9.34					
Bladder neck	0.47	0.65	0.2	2.12					
Non-papillary tumor	0.33	1.51	0.66	3.48	Non-papillary tumor	0.55	0.04	>0.01	1955.36
Multiple tumors	>0.01	3.04	1.56	5.9	Multiple tumors	0.5	0.46	0.05	4.39
Hematuria	0.7	1.14	0.59	2.17	Hematuria	0.99	0.98	0.14	7
Positive cytology	0.19	1.6	0.8	3.22	Positive cytology	0.21	4.28	0.44	41.43
BCG treatment	0.46	1.33	0.63	2.82	BCG treatment	0.82	1.31	0.14	12.63
PDD	0.88	1.12	0.27	4.69					

## Data Availability

The data presented in this study is available within the article.

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
