# Peer review of "Does Bladder Cancer with Inchworm Sign Indicate Better Prognosis after TURBT?"

_cancers, 2022, doi:10.3390/cancers14235767_

Round 1

Reviewer 1 Report

Inchworm sign is an MRI finding incorporated into Vi-RADS classification. While its predictive value for muscle invasion has been proven, almost no studies assessed its connection with cancer progression. Thus the topic of the manuscript is important and understudied now. In general the methodology of the study is proper, however, several aspects require major revision.

Q1. Introduction, line 47. «…locally advanced BC that has invaded the muscle layer, which is called muscle invasive bladder cancer (MIBC)…» The term locally advanced BC means cancer that spread to the paravesical tissues or to the regional lymph nodes (stage T3 or N1), while MIBC is a stage T2 disease. The authors are suggested to remove the term «locally advanced»

Q2. Please, consider explaining at the introduction in a few words the meaning of the «inchworm sign».

Q3. The authors are encouraged to provide a detailed inclusion and exclusion criteria. Did they include only primary tumors, or relapses as well? Were cases with multiple tumors included? Did they rule out UTUC?

Q4. Could you please provide the number of surgeons? Did all of them use similar technique and equipment? Did all the patients receive immediate postoperative intravesical chemotherapy?

Q5. Could you please add the data on the tumor size and number to the Table 3? These aspects may influence the outcomes, so it is important to see, whether the groups were comparable in them.

Author Response

Reply to the reviewer 1

Q1. Introduction, line 47. «…locally advanced BC that has invaded the muscle layer, which is called muscle invasive bladder cancer (MIBC)…» The term locally advanced BC means cancer that spread to the paravesical tissues or to the regional lymph nodes (stage T3 or N1), while MIBC is a stage T2 disease. The authors are suggested to remove the term «locally advanced»

Thank you for your comments. We agree with you about your mention. We have removed the relevant part (line 50).

Q2. Please, consider explaining at the introduction in a few words the meaning of the «inchworm sign».

Thank you for your comments. We have added a description of the imaging findings of the inchworm sign to the introduction (lines 65-67).

Q3. The authors are encouraged to provide a detailed inclusion and exclusion criteria. Did they include only primary tumors, or relapses as well? Were cases with multiple tumors included? Did they rule out UTUC?

Thank you for your comments. The present study included only initial cases and did not include recurrent cases. Cases with multiple tumors were included. We also excluded cases with coexisting UTUC at the initial diagnosis (lines 76-79).

Q4. Could you please provide the number of surgeons? Did all of them use similar technique and equipment? Did all the patients receive immediate postoperative intravesical chemotherapy?

Thank you for your comments. Patients treated from 2007 to 2019 were included. There was a total of 37 surgeons in the 12-year period, due to staff transitions at our facility. All surgeons were similar in terms of surgical techniques. For surgical instruments, Olympus TURis bipolar resectoscope (Olympus Corporation, Tokyo, Japan) was used, and Aladuck LS-DLED (SBI Pharma, Tokyo, Japan) or Storz PDD system and Storz bipolar resectoscope (KARL STORZ, Tuttlingen, Germany) was used for PDD-TURBT.

Twenty-one patients received immediate postoperative intravesical chemotherapy. We suspect that this variation was due to the fact that intravesical chemotherapy is not routinely administered at our facility, and the physician's discretion determines whether or not to administer the drug and what type of drug to use. The drugs administered varied from Pirarubicin, Mitomycin C, and Epirubicin Hydrochloride (lines 126-133).

Q5. Could you please add the data on the tumor size and number to the Table 3? These aspects may influence the outcomes, so it is important to see, whether the groups were comparable in them.

Thank you for pointing this out to us. Additional studies were investigated and found significant differences in tumor size between the two groups (p < 0.01). Huan-Jun et al. reported that the median diameter of tumors with stems (21.5 mm) was significantly larger than that of tumors without stems (13.0 mm)1. They also postulated that submucosal enhancement and stalks are related to the size of the tumors and small tumors may not generate enough pulling force to form stalks. In our study, this may be related to the significantly larger tumor size in NMIBC patients with inchworm sign. There was no significant difference in multiple tumors (lines 170-171, 246-251 and Table3).

  1. Wang HJ, Pui MH, Guan J, Li SR, Lin JH, Pan B, Guo Y. Comparison of Early Submucosal Enhancement and Tumor Stalk in Staging Bladder Urothelial Carcinoma. AJR Am J Roentgenol. 2016 Oct;207(4):797-803. doi: 10.2214/AJR.16.16283. Epub 2016 Aug 9. PMID: 27505309.

Reviewer 2 Report

<Summary>

This study aimed to evaluate the clinical relevance of inchworm sign in NMIBC treated with TURBT and/or intravesical therapy. The reviewer appreciates this work and effort to obtain the conclusion. There are still some of limitations to be improved before this work is accepted in this journal.

 <Comments>

1. The authors describe ‘positive urine cytology was an unfavorable factor for recurrence-free survival in patients with non-muscle invasive bladder cancer with inchworm sign.’

Generally speaking, positive cytology is a sign of high-grade tumor and aggressive tumor. In this sense, positive cytology is associated with an unfavorable factor not only in NMIBC with inchworm sigh. This is not a special finding. Do not exaggerate the result.

2. The authors concluded ‘The inchworm sign is a helpful finding in predicting NMIBC’. This is not a novel finding. Looking at VIRADS, the importance of inchworm sign has been already reported. What is the strength of this study?

3. Generally, MRI examination costs a lot. This cannot be a routine clinical practice. Do the authors recommend physicians to exam MRI before TURBT for all the patients with bladder tumor?

4. The inchworm sign is kind of subjective, which means the result may vary across the radiologist and urologists. What about the concordance between observers. Who decide positive or negative for the inchworm sign? The reviewer strongly recommend to add results of inter-observer variability and concordance index.

5. What is the point and reason of selecting the two cases in Figure 2? Patients with inchworm sign showed muscle invasion. Are they typical cases?

6. Table 5

Some of the factors showed super crazy data of upper 95%CI HR value. This is probably due to small sample size developing progression. The statistical analysis is problematic and did not make sense. The reviewer recommends to delete the data. How many patients developed progression during follow-up?

7. What is the mechanism underlying the formation of inchworm sigh? Add the detail in the discussion section.

Author Response

Reply to the reviewer 2

1.The authors describe ‘positive urine cytology was an unfavorable factor for recurrence-free survival in patients with non-muscle invasive bladder cancer with inchworm sign. Generally speaking, positive cytology is a sign of high-grade tumor and aggressive tumor. In this sense, positive cytology is associated with an unfavorable factor not only in NMIBC with inchworm sign. This is not a special finding. Do not exaggerate the result.

 Thank you for your comment. You are right that positive cytology is not a prognostic factor that applies only to NMIBC with inchworm sign, and we agree with you on this point. We have removed this result from the Summary, Absrtact and Conclusions as we do not believe it should be exaggerated (lines 19-20, 38-40, 299-300).

  1. The authors concluded ‘The inchworm sign is a helpful finding in predicting NMIBC’. This is not a novel finding. Looking at VIRADS, the importance of inchworm sign has been already reported. What is the strength of this study?

 Thank you for pointing this out to us. One of the most important findings of our study is the identification of bladder neck and papillary tumor as two predictors of MIBC in bladder cancer with inchworm sign. In other words, we would like to suggest that it should be noted that the presence of an inchworm sign was indeed a high probability of NMIBC, but this was not the case in all cases. As you pointed out, we wonder if our stated conclusions may not adequately convey the strengths of this study. We have revised some of our conclusions and summary (33-38, 296-297).

  1. Generally, MRI examination costs a lot. This cannot be a routine clinical practice. Do the authors recommend physicians to exam MRI before TURBT for all the patients with bladder tumor?

 Thank you for your comment. As you point out, we believe that MRI is never an essential examination as it is one of the more expensive tests. However, prior to performing TURBT, it is very important to predict muscle invasion by imaging studies, and MRI is a more effective tool than CT in assessing the degree of invasion. The results of our study do not provide evidence to recommend MRI for all patients with bladder cancer. To prove that MRI is necessary for all patients, it is needed to compare the prognosis and treatment efficacy between patients who underwent preoperative MRI and those who did not, for example. We believe that our study will assist in the diagnosis of bladder cancer depth by preoperative MRI.

  1. The inchworm sign is kind of subjective, which means the result may vary across the radiologist and urologists. What about the concordance between observers. Who decide positive or negative for the inchworm sign? The reviewer strongly recommends to add results of inter-observer variability and concordance index.

Thank you for your comments. The following additional examination of image variability has been performed.

A single urologist with 7 years of experience in reading abdominal DWI data and performing TURBT (Reviewer 1) reviewed the MR images blinded to clinical and histologic information. The tumor image with high signal intensity and a submucosal stalk with low signal intensity on DWI was identified as positive for the inchworm sign. To ensure interobserver concordance in assessment of the inchworm sign, one reviewer (Reviewer 2: a radiologist with 19 years of experience reading abdominal DWI data), who did not disclose the clinical course and histological findings, independently evaluated the presence of inchworm signs on all images (lines 92-101).

We assessed interobserver variability in the identification of inchworm signs. Reviewers 1 and 2 independently analyzed DW images of all eligible tumors and found inchworm signs in 108 (33.2%) and 112 (34.5%) tumors, respectively. Interobserver variability between the two readers was generally excellent with a kappa coefficient of 0.85 (lines 199-203).

  1. What is the point and reason of selecting the two cases in Figure 2? Patients with inchworm sign showed muscle invasion. Are they typical cases?

 Thank you for your comments. These are two cases of MIBC despite bladder cancer with inchworm sign. In the first case, we have picked up images of a patient with a non-papillary tumor, and in the second case, we have picked up images of a patient with a tumor located in the bladder neck. These two factors are predictors of MIBC in patients with inchworm sign, as identified by Cox analysis, and images were taken from patients in this study as examples of MIBC patients with these factors. We believe this figure is very important as it gives a strong impression of the predictors in MIBC that have been identified.

  1. Table 5

Some of the factors showed super crazy data of upper 95%CI HR value. This is probably due to small sample size developing progression. The statistical analysis is problematic and did not make sense. The reviewer recommends to delete the data. How many patients developed progression during follow-up?

 Thank you for your comments. Of the NMIBC patients with negative Inchworm sign, only 4 (5.2%) of the 77 patients progressed. As you point out, I think the small sample size as well as the small number of events is what causes the 95% CI values to be aberrant. Items with a 95% CI Upper greater than 10000 have been removed (Table5).

  1. What is the mechanism underlying the formation of inchworm sign? Add the detail in the discussion section.

Thank you for your comments. Saito et al. reported that the stalk extending from the bladder wall to the tumor center is composed of fibrous tissue, capillaries, inflammatory cells, and edema1. Takeuchi et al. also reported that pT1 bladder cancer with inchworm sign histologically has a submucosal stalk consisting of significantly edematous submucosa, fibrous tissue, capillaries, and mild inflammatory cell infiltration, which is consistent with a low signal intensity region on DW-MRI2. Although there are no reports on the formation of the inchworm sign, we suspect that the thickening of the submucosa due to inflammation and edema results in the formation of this bow-shaped structure (lines 209-221).

  1. Saito W, Amanuma M, Tanaka J, Heshiki A. Histopathological analysis of a bladder cancer stalk observed on MRI. Magn Reson Imaging. 2000 May;18(4):411-5. doi: 10.1016/s0730-725x(00)00124-7. PMID: 10788718.
  2. Takeuchi M, Sasaki S, Ito M, Okada S, Takahashi S, Kawai T, Suzuki K, Oshima H, Hara M, Shibamoto Y. Urinary bladder cancer: diffusion-weighted MR imaging--accuracy for diagnosing T stage and estimating histologic grade. Radiology. 2009 Apr;251(1):112-21. doi: 10.1148/radiol.2511080873. PMID: 19332849.

Round 2

Reviewer 2 Report

The authors have responded to the reviwer's comments, suggestions, and recommendations, properly.

Author Response

Thank you very much.